# A Novel Integrated APCI and MPT Ionization Technique as Online Sensor for Trace Pesticides Detection

**DOI:** 10.3390/s22051816

**Published:** 2022-02-25

**Authors:** Gaosheng Zhao, Fengjian Chu, Jianguang Zhou

**Affiliations:** 1State Key Laboratory of Industrial Control Technology, Institute of Cyber-Systems and Control, Research Center for Analytical Instrumentation, Zhejiang University, Hangzhou 310027, China; zgs571@163.com; 2College of Information Science and Electronic Engineering, Zhejiang University, Hangzhou 310027, China; 22031043@zju.edu.cn; 3Department of Chemistry, Zhejiang University, Hangzhou 310027, China

**Keywords:** online sensor, portable, ambient mass spectrometry, pesticide detection, high sensitivity, multi-mode ion source

## Abstract

The misuse of pesticides poses a tremendous threat to human health. Excessive pesticide residues have been shown to cause many diseases. Many sensor detection methods have been developed, but most of them suffer from problems such as slow detection speed or narrow detection range. So, the development of rapid, direct and sensitive means of detecting trace amounts of pesticide residues is always necessary. A novel online sensor technique was developed for direct analysis of pesticides in complex matrices with no sample pretreatment. The portable sensor ion source consists of an MPT (microwave plasma torch) with desolventizing capability and an APCI (atmosphere pressure chemical ionization), which provides abundant precursor ions and a strong electric field. The performance which improves the ionization efficiency and suppresses the background signal was verified by using pesticide standard solution and pesticide pear juice solution measurements with an Orbitrap mass spectrometer. The limit of detection (LOD) and the limit of quantization (LOQ) of the method were measured by pear juice solutions that were obtained in the ranges of 0.034–0.79 μg/L and 0.14–1 μg/L. Quantitative curves were obtained ranging from 0.5 to 100 μg/L that showed excellent semi-quantitative ability with correlation coefficients of 0.985–0.997. The recoveries (%) of atrazine, imidacloprid, dimethoate, profenofos, chlorpyrifos, and dichlorvos were 96.6%, 112.7%, 88.1%, 85.5%, 89.2%, and 101.9% with the RSDs ranging from 5.89–14.87%, respectively. The results show that the method has excellent sensitivity and quantification capability for rapid and direct detection of trace pesticide.

## 1. Introduction

As a new sensor technology, mass spectrometry has been widely used in biology, medicine and the environment, etc. Laser mass spectrometry as an online sensor has great potential to be applied to industrial processes such as for the analysis of coffee roasting [1], and electrospray ionization mass spectrometry (ESI-MS) has been coupled with microdialysis sampling to perform in vivo chemical monitoring. The MS “sensor” has been used to monitor acetylcholine in the brain of live rats [2]. The imaging mass-spectrometry sensor is a powerful technique that allows for storing multiple time stamps in each pixel for each time-of-flight cycle to obtain more chemical information on a sample [3]. PTR-MS as a multipurpose sensor has also been widely used in environmental, agri-food and health science sectors [4]. The development of new mass-spectrometry detection technology has great significance to expanding sensor application fields, such as high-sensitivity and rapid detection of complex substances.

Pesticides have been widely used in agricultural production to kill insects, fungi, and other organisms that are harmful to the growth and yield of crops. However, different levels of pesticide residues in fruits, vegetables, as well as in the environment, pose an immense threat to human health. Abundant evidence has shown a correlation between pesticides and the development of a wide spectrum of pathologies, ranging from eczema to neurological disorders and cancer [5]. Meanwhile, due to bioaccumulation, even small amounts of residues can be harmful to humans through food. Therefore, various countries have established indicators for maximum residue levels (MRLs) of these compounds in food and drinking water to help protect the community against contamination and potential adverse health effects. In the new stringent directives of the European Union (EU), the MRLs for pesticides in fruits and vegetables range from 10s and 100s of ppb (μg/kg) to ppm (mg/kg) [6].

Sensors for pesticides’ detection are urgently required to control food safety, including traditional pesticide optical sensors such as fluorescence (FL), colorimetric (CL), surface-enhanced Raman scattering (SERS), surface plasmon resonance (SPR) sensors, which provide comprehensive coverage of pesticide detection with high sensitivity [7,8,9,10]. Although these techniques offer a simple structure and operation, they also have many drawbacks such as few detection material types and poor detection accuracy. Thus, to develop rapid detection of pesticide residues, a variety of analytical techniques have been used. Gas chromatography/mass spectrometry (GC-MS) and high-performance liquid chromatography/mass spectrometry (HPLC-MS) remain the preferred traditional multi-residue analytical methods for determining pesticide residues [11]. However, chromatography-mass spectrometry techniques are time-consuming, and the pretreatment process is laborious. In recent years, with the development of ambient MS, various atmospheric pressure ionization sources have been used for pesticide testing, including desorption electrospray ionization (DESI) [12,13,14], extractive electrospray ionization (EESI) [15,16], paper spray ionization (PSI) [17,18,19,20], probe electrospray ionization [21], and atmospheric plasma ionization sources, such as direct analysis on real time (DART) [22,23,24,25], atmospheric pressure chemical ionization (APCI) [26,27], and dielectric barrier discharge ionization (DBDI) [28,29,30,31]. These ion sources avoid complex pretreatment and chromatographic separation steps and offer multiple advantages. In addition, the microwave plasma torch (MPT) is a recently developed microwave plasma-based ambient ionization source with the advantages of miniaturization, excellent ionization efficiency, and strong excitation ability [32].

A number of multi-mode ambient ion-generation techniques have been previously reported, including APCI/ESI (electrospray ion source [33], DESI/metastable-induced ionization [34], interactive mass spectrometry imaging (IMSI) [35], and integrated ambient ionization source (iAmIS) [36]. These techniques compensate for features such as analytical polarity and mass range, provide flexibility in choosing different ionization modes, broaden the scope of analyte detection, and facilitate the analysis of complex samples.

In this study, we present a sensitive, portable and direct multi-mode ionization technique as an online sensor that integrates APCI and MPT sources with an Orbitrap mass spectrometer. The combination of APCI and MPT has many advantages. The low ionization efficiency of MPT for some low-polarity organics can be well-enhanced by the APCI source, while the high electric field in APCI can drive more ions from the atmospheric MPT ion source into the mass spectrometer. The improved ionization efficiency of this technique enhances the possibility of directly analyzing the ultra-low concentration of pesticide residues. To validate the performance of this dual-source method, a range of common pesticides were tested in three ionization modes (APCI-only, MPT-only, MPT-APCI) and evaluated for sensitivity, desolventizing effects, and ion fragmentation. The integrated APCI and MPT ion source technology exhibits impressive detection limits at several ppt levels, which exceed most current atmospheric pressure ion sources, and has proven to be a rapid and sensitive analytical method for trace pesticide residues without pretreatment.

## 2. Materials and Methods

Atrazine and imidacloprid were purchased from Yuanye Bio-Technology Co., Ltd. (Shanghai, China); dimethoate, profenofos, and chlorpyrifos were purchased from Xianding Biotechnology Co., Ltd. (Shanghai, China); dichlorvos was purchased from J&K Scientific Co., Ltd. (Shanghai, China). Methanol was purchased from Merck Co. (Darmstadt, Germany). Ar (99.999% purity) was purchased from Shanghai Ji Liang Standard Reference Gases Co., Ltd. (Shanghai, China). We purchased the pear from a local supermarket in Hangzhou, China. The juice was squeezed directly after rinsing the surface of the pear, then it was left to stand for 3 min to obtain pear juice from the liquid supernatant.

The solid pesticide samples were stored in a refrigerator at 4 °C. Pesticide standard solutions were prepared with methanol at concentrations of 10, 1 and 0.1 0.01 μg/mL, and a concentration gradient of 100, 50, 10, 5, 1, 0.5, 0.1 μg/L of standard pesticide solutions were prepared with pear juice. A quartz capillary is dipped into the measured sample and placed at the tip of the plasma torch, where the high-temperature plasma desorbs the sample and produces gas phase molecules/ions in the corona region. The quartz capillary with a diameter of 1 mm was used and the dipping depth of the sample solution was 10 mm. However, differences in surface tension of different concentrations solutions can cause deviations in the amount of absorption. Each injection volume is 5–10 µL.

### 2.1. The Multi-Mode Ion Source

A novel multi-mode ion source was constructed by combining an MPT source with an APCI source. The dual ion sources were placed orthogonally, as shown in Figure 1, with the distance from the plasma torch to the APCI corona needle being equal to the distance from the needle to the mass analyzer inlet (D1 = D2 = 5 mm). When D1 > 5 mm, the two ionization sources do not integrate well in terms of ionization of analytes. When D2 > 10 mm, fewer ions were desorbed into the mass spectrometer. The MPT source consists of three concentric tubes, namely, the microwave input tube (outer tube), the intermediate tube, and the argon flow tube (inner tube). The outer tube (22 mm O.D., 11.5 mm I.D.) and the inner tube (1.5 mm O.D., 0.8 mm I.D.) are made of copper. The central tube (5 mm O.D., 2.8 mm I.D.) is made of brass. Argon gas was introduced from the inner and central tubes to generate plasma and to modify the plasma jet shape; the flow rate of argon gas was 1000 mL/min for both tubes and the argon gas could be adjusted by two rotameters (HORIBA METRON, Beijing, China). Microwave signals are added to the intermediate tube via a coaxial cable. The microwave generator has a maximum power of 100 W and a frequency of 2.45 GHz. The APCI needle is secured to the front of the MPT by a 3D printed carabiner, on which a 3000 V DC voltage is applied using a homemade power supply.

The entire unit was mounted on an xyz three-axis moving platform and controlled by two separate power supplies, thus providing three operating modes: APCI-only, MPT-only and APCI-MPT modes. The three modes can be switched quickly and conveniently via two switches.

### 2.2. Mass Spectrometer

All MS experiments were performed using an LTQ Orbitrap mass spectrometer (Thermo Fisher Scientific, Waltham, MA, USA). The basic operating conditions were established as follows: capillary voltage: 30 V; capillary temperature: 300 °C; tube lens voltage: 100 V. Peak integration and data acquisition were performed via the instrument-embedded Xcalibur^®^ software (version 2.2 SP1.48; TFS, San Jose, CA, USA). The time scale of each scan is normally 100 ms under experimental conditions and can be varied according to specific requirements. All experimental data were obtained in positive ion mode. Prior to each sample measurement, the precise mass analysis was calibrated using the following ESI parameters: sheath gas (nitrogen) flow rate, 6 arb; aux gas (nitrogen) flow rate, 0 arb; capillary temperature, 275 °C; spray voltage, 5 kV; capillary voltage, 35 V; tube lens voltage, 110 V.

## 3. Results and Discussion

### 3.1. Analysis of the Novel Integrated APCI and MPT Ion Source

Both MPT and APCI, as ambient mass spectrometry ion sources, are exposed to the atmosphere during operation. Both sources generate ions that have certain effects on the analysis of samples, such as high-energy electron bombardment and proton transfer reaction, which may promote the identification of samples [37]. MPT generated abundant high-energy electrons, cations, active free radicals, and metastable particles with thermal and luminous energies, leading to desolvation and charge transfer from the solvent to the target, the detailed mechanism of which has been reported in a previous study [38]. The APCI source generated abundant precursor ions in the reaction zone, including [H3O]+ and [O]−, which usually undergo a charge transfer reaction with sample compounds. Table 1 summarizes the basic information of these pesticides in MPT-APCI mode (methanol solutions) of different pesticides with concentrations ranging from 0.01–10 μg/mL. [M + H]+ was detected by the APCI and MPT ionization sources. It is worth noting that the ion species detected by the three ionization modes are the same.

For integrated APCI and MPT ionization, the high-temperature airflow and electric field within the source are believed to be the main factors in improving detection sensitivity. For the MPT source, the sample molecules were ionized and then pushed into the mass spectrometry inlet mainly through argon as the carrier gas, while in the APCI source, the ions enter the transport tube spectrometry inlet mainly via the electric field and the suction force of the mass. In the integrated APCI and MPT sources, the ions are guided by the electric and flow fields, which improve the transport efficiency with a consequent increase in sensitivity.

The novel integrated APCI and MPT ion source improves the ionization efficiency of the target analytes and suppresses the background signal. Figure 2 shows the mass spectra of atrazine (10 μg/mL) and dichlorvos (1 μg/mL) in three ionization modes. The protonated atrazine appears at *m*/*z* 216.101, with the typical isotope of chlorine occurring at *m*/*z* 218.098; 174.054 is the characteristic fragment ion of atrazine. *m*/*z* 220.953 is the protonated dichlorvos, with the typical isotope occurring at *m*/*z* 222.951. For atrazine, the background mass peak abundance was prominent and even exceeded the target mass signal in the APCI-only mode, while in the MPT-only and MPT-APCI modes, the background signal was barely invisible in the *m*/*z* range of 50–600. This may be due to the fact that the new ion source operates in a pure working gas, and impurities in the matrix are reduced by airflow and thermal desorption. A similar effect was observed for dichlorvos (1 μg/mL). The dual source remarkably increased the abundance of the target ions. This effect is attributed to the high-temperature thermal desorption of MPT and was measured at a temperature of 3000 K using thermocouples in the MPT-generated plasma torch.

In addition, the metastable Ar produced by MPT carries high energy and dramatically enhances ionization efficiency through Penning ionization. Meanwhile, the microwave plasma torch has a high temperature and a pure working gas, which can enhance the sample dissolution effect so as to increase the probability of ionization and reduce the background interference. The precursor ions produced by APCI ionized air are also considered to be one of the critical factors to improve the sensitivity of the ion source. The results mechanistically reflect the reason for the increased sensitivity of the neighborhood activated ion source and provide ideas for the optimization of other similar atmospheric ion sources.

### 3.2. Real Complex Matrices Sample Analysis

To verify the capability of the online sensor analytical method for complex matrices, we take pear juice with added atrazine, imidacloprid, dimethoate, profenofos, chlorpyrifos, and dichlorvos as internal standards. To demonstrate the semi-quantitative capability of the source, the ions used for the qualitative analysis of atrazine, imidacloprid, dimethoate, profenofos, chlorpyrifos, and dichlorvos had *m*/*z* of 216.10, 256.06, 230.01, 374.94, 349.93, and 220.95, respectively. Figure 3 shows the linear responses of (A) the profenofos analysis using *m*/*z* 374.94 and chlorpyrifos using *m*/*z* 349.93 in the 100, 50, 10, 5, and 1 μg/L range; and (B) the dimethoate, atrazine, dichlorvos, and imidacloprid analyses in the 100, 50, 10, 5 and 0.5 μg/L range in pear juice. The calibration curves of the different pesticides possessed high linearity for concentrations over a range of more than two orders of magnitudes with an R2 between 0.985 and 0.997. Table 2 shows the results of the actual measurements of pesticides in fruit juices, including linearities and recoveries, and the recoveries (%) of atrazine, imidacloprid, dimethoate, profenofos, chlorpyrifos, and dichlorvos were 96.6%, 112.7%, 88.1%, 85.5%, 89.2%, and 101.9%, respectively. The instrumental limit of detection (LOD) and the limit of quantitation (LOQ) for all analytes were calculated based on a signal-to-noise ratio (S/N) of 3 and 10. The LOD and LOQ for each pesticide compound were measured ranging from 0.034 to 0.79 μg/L and 0.14–1 μg/L with a relative standard deviation (RSD, *n* = 6) less than 14.87%. The atmospheric plasma usually has high RSD, hence the RSD seems to be relatively slightly high. It is evident that the dual source provides better performance for different types of pesticides; the detection limit for atrazine reached 34 ng/L, exceeding most previous ionization methods. Table 3 shows the performance of the different ion source techniques used for pesticide analysis. The detection sensitivity of DBD and photoionization is worse than the APCI-MPT ionization techniques for trace pesticides detection. The PSI, PESI, and DART have proximity performance, but the DART ionization techniques need the methanolic extraction preparation with the PSI and PESI, which have complex background signals resulting in a difficult spectrum analysis. It can be seen that the novel ion source attains or exceeds the existing ionization techniques and can be used as a new choice for the rapid detection of pesticide residues in a solution.

## 4. Conclusions

In this study, a novel multi-mode ion source with an Orbitrap mass spectrometer as online sensor has been used to rapidly ionize a series of pesticides, including atrazine, imidacloprid, dimethoate, profenofos, chlorpyrifos, dichlorvos, and paraquat. The dual ion source can be operated in MPT-only, APCI-only and MPT-APCI modes by adjusting the power supply at the source. Combining the thermal desorption of MPT with the strong electric field of APCI, this method exhibits good desolventizing ability and high sensitivity. The LODs in the MPT-APCI mode ranged from 0.034–0.79 μg/L under optimal conditions using a complex matrices solution of the analytes. The method also showed excellent semi-quantitative ability with correlation coefficients of 0.985–0.997 in the range of 0.5–100 μg/L. This online sensor method is very rapid, simple and efficient for determining low concentrations of pesticides for online analysis without any pretreatment. Additionally, the technology that integrates APCI and MPT sources with a mass spectrometer can be subsequently miniaturized and loaded on a mobile detection vehicle, so that it can be driven to the site of detection at any time for online inspection. In general, this aspect of enhanced sensitivity is expected to be widely applied to plasma sources and may provide ideas for the direct detection of trace substances in complex matrices.

## Figures and Tables

**Figure 1 sensors-22-01816-f001:**
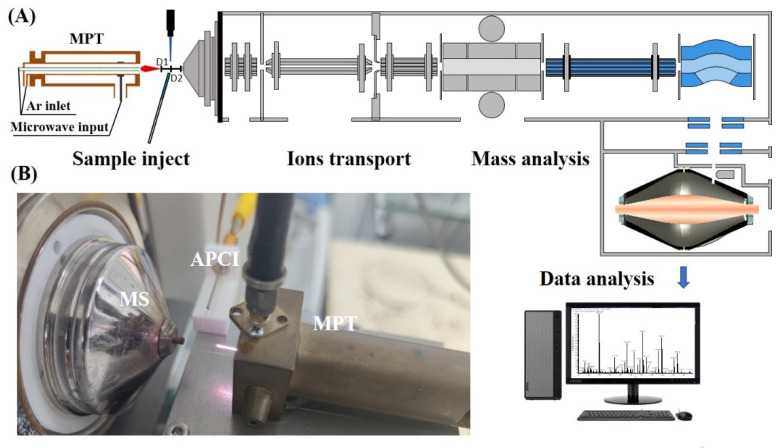
(**A**) Schematic diagram of the online sensor structure. (**B**) Physical view of the ion source.

**Figure 2 sensors-22-01816-f002:**
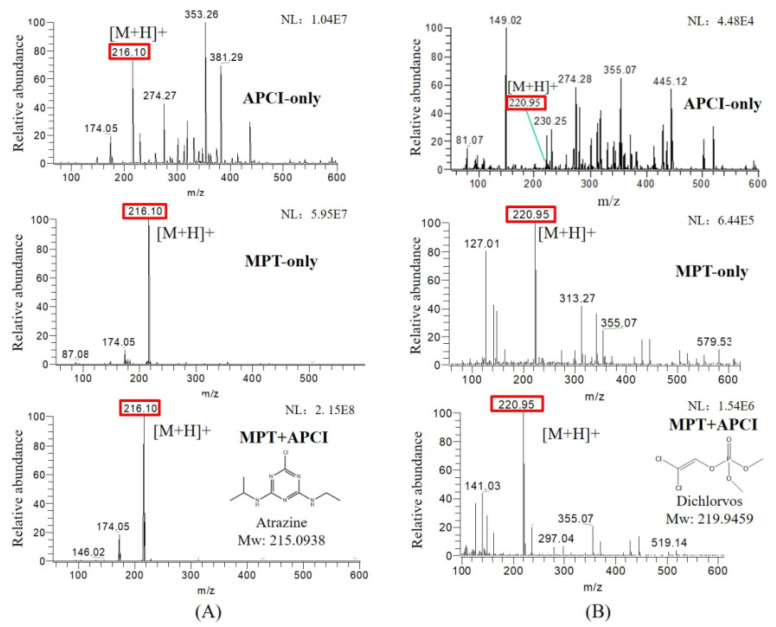
Mass spectra of methanol solutions of (**A**) 10 μg/mL atrazine and (**B**) 1 μg/mL dichlorvos in APCI-only, MPT-only and MPT-APCI modes, NL represents signal strength in the figure.

**Figure 3 sensors-22-01816-f003:**
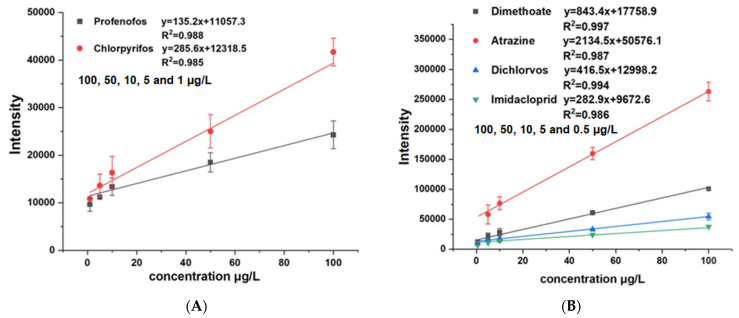
The linear responses of (**A**) the profenofos analysis using *m*/*z* 374.94 and chlorpyrifos using *m*/*z* 349.93 in the 100, 50, 10, 5 and 1 μg/L range; and (**B**) the dimethoate, atrazine, dichiorvos and imidacloprid analyses in the 100, 50, 10, 5 and 0.5 μg/L range in pear juice.

**Table 1 sensors-22-01816-t001:** Characteristics of the six pesticides used in the study from the results of MS analysis of the standard pesticide samples.

Compound	Molecular Formula	MW (*m*/*z*)	Ion Form	Molecular Structure
Atrazine	C_8_H_14_ClN_5_	215.09 217.09	[M + H]^+^	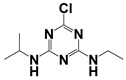
Imidacloprid	C_9_H_10_ClN_5_O_2_	255.05 257.05	[M + H]^+^	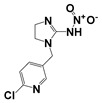
Dimethoate	C_5_H_12_NO_3_PS_2_	229.00	[M + H]^+^	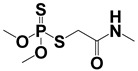
Profenofos	C_11_H_15_BrClO_3_PS	371.93 373.93	[M + H]^+^	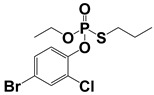
Chlorpyrifos	C_9_H_11_Cl_3_NO_3_PS	348.92 350.92	[M + H]^+^	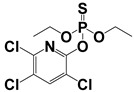
Dichlorvos	C_4_H_7_Cl_2_O_4_P	219.94 221.94	[M + H]^+^	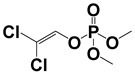

**Table 2 sensors-22-01816-t002:** Results for the analysis of the correlation coefficient (R2) of pesticides in pear juice, linear range, recovery rate (%), and limit of detection/quantitation (LOD and LOQ) of the method.

Pesticides	Equation	*R* ^2^	Linear Range (μg/L)	RSD (%)	Recovery Rate * (%)	LOD (g/L)	LOQ (μg/L)
Profenofos	*y* = 135.2*x* + 11057.3	0.988	1–100	10.5	85.5	0.74	0.98
Chlorpyrifos	*y* = 285.6*x* + 12318.5	0.985	1–100	14.87	89.2	0.79	1.00
Dimethoate	*y* = 843.4*x* + 17758.9	0.997	0.5–100	13.07	88.1	0.17	0.43
Atrazine	*y* = 2134.5*x* + 50576.1	0.987	0.5–100	5.89	96.6	0.034	0.14
Dichiorvos	*y* = 416.5*x* +12998.2	0.994	0.5–100	10.2	101.9	0.085	0.34
Imidacloprid	*y* = 282.9*x* + 9672.6	0.986	0.5–100	8.74	112.7	0.077	0.26

* Using the concentration of 2.5 μg/L.

**Table 3 sensors-22-01816-t003:** Performance of different ion source techniques for pesticide analysis.

Technology	Sample Preparation	MS Analysis	Analytical Performance	Ref.
PSI	Not required	Orbitrap-MS	LOD < 1.25 pmol	[39]
PESI ^a^	Not required	TOF-MS ^b^	LOD < 50 pg	[40]
DART	IT-SPME ^c^	TOF-MS	LOQs: 0.06–0.46 μg L^−1^	[41]
DBD	Methanolic extraction	QqQ-MS	LODs: 10–1000 μg L^−1^	[42]
Photoionization	Not required	IT-MS ^d^	LODs: 0.1–1 μg L^−1^	[43]
APCI-MPT	Not required	Orbitrap-MS	LODs: 0.034–0.79 μg L^−1^	-

^a^ Probe electrospray ionization. ^b^ In-tube solid-phase microextraction. ^c^ Time-of-flight mass spectrometry. ^d^ ion trap mass spectrometry.

## Data Availability

Not applicable.

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
