# Peer review of "A Novel Integrated APCI and MPT Ionization Technique as Online Sensor for Trace Pesticides Detection"

_sensors, 2022, doi:10.3390/s22051816_

Round 1

Reviewer 1 Report

The concept is interesting and brings value to the sensors technologies. Some critical discussion of the results could increase the scientific soundness of the manuscript. In parts, the results section feels like loose collections of statements without little comments or assessment. Hence, a comparison with the reported studies in table 3 should be made. 

Author Response

The concept is interesting and brings value to the sensors technologies. Some critical discussion of the results could increase the scientific soundness of the manuscript. In parts, the results section feels like loose collections of statements without little comments or assessment. Hence, a comparison with the reported studies in table 3 should be made. 

Point 1: The concept is interesting and brings value to the sensors technologies. Some critical discussion of the results could increase the scientific soundness of the manuscript. In parts, the results section feels like loose collections of statements without little comments or assessment. Hence, a comparison with the reported studies in table 3 should be made. 

Response 1: Thank you very much for the suggestion. The words in the results section"The LOD and LOQ for each pesticide compound were measured range from 0.034 to 0.79μg/L and 0.14-1μg/L with a relative standard deviation (RSD, n=6) less than 14.87%.  It is evident that the dual source provides better performance for different types of pesticides; the detection limit for atrazine reached 34 ng/L, exceeding most previous ionization methods. Table 3 shows the performance of the different ion source techniques used for pesticide analysis." have been replaced by "The LOD and LOQ for each pesticide compound were measured range from 0.034 to 0.79μg/L and 0.14-1μg/L with a relative standard deviation (RSD, n=6) less than 14.87%. The atmospheric plasma usually has high RSD, hence the RSD seems to be a little high relatively. It is evident that the dual source provides better performance for different types of pesticides; the detection limit for atrazine reached 34 ng/L, exceeding most previous ionization methods. Table 3 shows the performance of the different ion source techniques used for pesticide analysis. The detection sensitivity of DBD and Photoionization is worse than the APCI-MPT ionization techniques for trace pesticides detection. The PSI,PESI, and DART have proximity performance, but the DART ionization techniques need the methanolic extraction preparation with the PSI and PESI have the complex background signals result in the difficulty of spectrum analysis.".

Reviewer 2 Report

In this paper the authors propose a multi-mode ionization technique for the online monitoring of pesticides. The proposed technique integrates APCI and MPT sources with an Orbitrap mass spectrometer in order to take advantage of the high electric field of the APCI to drive more ions from the atmospheric MPT ion source into the mass spectrometry. The system can operate in different modes and shows good performances in terms of sensitivity and desolventizing ability.

The work is well organized although it is full of typos, please correct it and pay attention  to the spaces between values and units and spaces between worlds in general.

The reviewer has the following concerns:

-It could be interesting to clarify how the developed technique  could be used for real time monitoring or in field measurements for the pesticides detection and monitoring. Could be possible to integrate it in a as compact as possible device?

- The authors assert that, by using these integrated techniques it is possible not only to improve the ionization efficiency of the target analyte but also it helps in the background signal suppression. Please, clarify this aspect.

Author Response

Point 1:In this paper the authors propose a multi-mode ionization technique for the online monitoring of pesticides. The proposed technique integrates APCI and MPT sources with an Orbitrap mass spectrometer in order to take advantage of the high electric field of the APCI to drive more ions from the atmospheric MPT ion source into the mass spectrometry. The system can operate in different modes and shows good performances in terms of sensitivity and desolventizing ability.

The work is well organized although it is full of typos, please correct it and pay attention to the spaces between values and units and spaces between worlds in general.

Response 1: Thank you very much for the suggestion. The spaces between values and units and spaces between worlds had been corrected.

The reviewer has the following concerns:

Point 2: It could be interesting to clarify how the developed technique could be used for real time monitoring or in field measurements for the pesticides detection and monitoring. Could be possible to integrate it in a as compact as possible device?

Response 2:Thank you a lot for the suggestion.We had added the words “Also, the technologythat integrates APCI and MPT sources with a mass spectrometercan be subsequently miniaturized, loaded on a mobile detection vehicle, so that it can be driven to the site of detection at any time for on-lineinspection.” in the paper.

Point 3: The authors assert that, by using these integrated techniques it is possible not only to improve the ionization efficiency of the target analyte but also it helps in the background signal suppression. Please, clarify this aspect.

Response 3:Thank you a lot for the suggestion.We had added the words “This may be due to the fact that the new ion source operates in a pure working gas, and impurities in the matrix are reduced by airflow and thermal desorption.” in the paper.

Reviewer 3 Report

The study “A novel integrated APCI and MPT ionization technique as on-line sensor for trace pesticides detection” describe a novel multi-mode ion source with an Orbitrap mass spectrometer as online sensor used to rapidly ionize a series of pesticides, including atrazine, imidacloprid, dimethoate, profenofos, chlorpyrifos, dichlorvos, and paraquat. Although the MS is well structured and show novelty but authors should reworked on some points. Therefore, we recommend this manuscript to proceed with MINOR REVISION. Main concerns are listed as follows:

  1. Author need to discuss about the market value of this type of sensor? What is the future of device in terms of commercialization and their economic value?
  2. There are extensive researches carried out for the development of distinct sensing platform like electrochemical, microfluidic etc. These sensors are sensitive, rapid, and low cost. How you justify your sensor as better than these in terms of cost.
  3. What was the sensitivity of developed sensor? Did authors calculate it?
  4. Data is not enough to support the claim. Add some more illustrations about the process of constructing sensor.
  5. Figure’s captions should be rewrite.
  6. Some typo mistakes are present in the manuscript. Please remove them.
  7. Authors mentioned the time scale of each scan is normally 100 ms under experimental conditions. What is the total expected time for complete detection including all experimental parameters used by them.
  8. Author should include some more relevant references. Some references are given below-

https://doi.org/10.1016/j.ab.2012.06.025, https://doi.org/10.1016/j.electacta.2012.02.012

https://doi.org/10.1016/j.ijbiomac.2020.08.215, https://doi.org/10.1016/j.procbio.2021.06.015

https://doi.org/10.1016/j.foodchem.2021.131126

Author Response

The study “A novel integrated APCI and MPT ionization technique as on-line sensor for trace pesticides detection” describe a novel multi-mode ion source with an Orbitrap mass spectrometer as online sensor used to rapidly ionize a series of pesticides, including atrazine, imidacloprid, dimethoate, profenofos, chlorpyrifos, dichlorvos, and paraquat. Although the MS is well structured and show novelty but authors should reworked on some points. Therefore, we recommend this manuscript to proceed with MINOR REVISION. Main concerns are listed as follows:

Point 1: Author need to discuss about the market value of this type of sensor? What is the future of device in terms of commercialization and their economic value?

Response 1:Thank you very much for the suggestion. The main application direction of this technologyis rapid detection on site. We had added the words “Also, the technologythat integrates APCI and MPT sources with a mass spectrometer can be subsequently miniaturized, loaded on a mobile detection vehicle, so that it can be driven to the site of detection at any time for on-lineinspection.” in the paper.

Point 2: There are extensive researches carried out for the development of distinct sensing platform like electrochemical, microfluidic etc. These sensors are sensitive, rapid, and low cost. How you justify your sensor as better than these in terms of cost.

Response 2: Thank you a lot for the suggestion. The new ion source technology can simultaneously detect multiple pesticides, giving accurate qualitative and quantitative results. Therefore, the mass spectrometer can be miniaturized by MEMS (Micro-Electro-Mechanical System) technology, which can reduce manufacturing and maintenance costs.

Point 3: What was the sensitivity of developed sensor? Did authors calculate it?

Response 3: The instrumental limit of detection (LOD) was calculated based on a signal to noise ratio (S/N) of 3. The LODsofthe ionization techniqueranged from 0.034-0.79μg/L underoptimal conditions using a complex matrices solution of the analytes.

Point 4: Data is not enough to support the claim. Add some more illustrations about the process of constructing sensor.

Response 4: Thank you a lot for the suggestion.We had added the picture about the process of constructing sensor.

Point 5: Figure’s captions should be rewrite.

Response 5: Thank you a lot for the suggestion. The figure’s captions had been rewritten.

Point 6: Some typo mistakes are present in the manuscript. Please remove them.

Response 6: Thank you a lot for the suggestion.The typo mistakes had been removed.

Point 7: Authors mentioned the time scale of each scan is normally 100 ms under experimental conditions. What is the total expected time for complete detection including all experimental parameters used by them.

Response 7: Thank you a lot for the suggestion.The whole detection time includes sampling and obtaining the detection result is less than 10 seconds;

Point 8: Author should include some more relevant references. Some references are given below-

Response 8: Thank you a lot for the suggestion.The relevant references had been added.